# A New Modeling Framework for Geothermal Operational Optimization with Machine Learning (GOOML)

**Grant Buster [1], Paul Siratovich [2,*], Nicole Taverna [1], Michael Rossol [1], Jon Weers [1], Andrea Blair [2], Jay Huggins [1], Christine Siega [3], Warren Mannington [3], Alex Urgel [3], Jonathan Cen [3], Jaime Quinao [4], Robbie Watt [4] and John Akerley [5]**

1   National Renewable Energy Laboratory (NREL), Golden, CO 80401, USA; grant.buster@nrel.gov (G.B.); nicole.taverna@nrel.gov (N.T.); Michael.Rossol@nrel.gov (M.R.); Jon.Weers@nrel.gov (J.W.); jay.huggins@nrel.gov (J.H.)
2   Upflow Limited, Taupo 3330, New Zealand; andy.blair@upflow.nz
3   Contact Energy Limited, Wairakei 3352, New Zealand; christine.siega@contactenergy.co.nz (C.S.); warren.mannington@contactenergy.co.nz (W.M.); alex.urgel@contactenergy.co.nz (A.U.); jonathan.cen@contactenergy.co.nz (J.C.)
4   Ngati Tuwharetoa Geothermal Assets Limited, Kawerau 3169, New Zealand; jaime.quinao@tuwharetoakawerau.co.nz (J.Q.); robbie.watt@tuwharetoakawerau.co.nz (R.W.)
5   Ormat Technologies Inc., Reno, NV 89519, USA; jakerley@ormat.com
*   Correspondence: paul.siratovich@upflow.nz

**Abstract:** Geothermal power plants are excellent resources for providing low carbon electricity generation with high reliability. However, many geothermal power plants could realize significant improvements in operational efficiency from the application of improved modeling software. Increased integration of digital twins into geothermal operations will not only enable engineers to better understand the complex interplay of components in larger systems but will also enable enhanced exploration of the operational space with the recent advances in artificial intelligence (AI) and machine learning (ML) tools. Such innovations in geothermal operational analysis have been deterred by several challenges, most notably, the challenge in applying idealized thermodynamic models to imperfect as-built systems with constant degradation of nominal performance. This paper presents GOOML: a new framework for Geothermal Operational Optimization with Machine Learning. By taking a hybrid data-driven thermodynamics approach, GOOML is able to accurately model the real-world performance characteristics of as-built geothermal systems. Further, GOOML can be readily integrated into the larger AI and ML ecosystem for true state-of-the-art optimization. This modeling framework has already been applied to several geothermal power plants and has provided reasonably accurate results in all cases. Therefore, we expect that the GOOML framework can be applied to any geothermal power plant around the world.

**Keywords:** geothermal power plant; systems modeling; machine learning; neural networks; system optimization; digital twins

## 1. Introduction

Geothermal energy production is the only renewable energy source that is not reliant on ambient or historical weather conditions to produce electricity reliably. Through harnessing the heat from underground reservoirs, operators of geothermal power plants use networks of steam and liquid gathering systems to provide motive fluids to turbines, yielding conditions that enable geothermal as a baseload energy source [1]. Opportunities for optimization of geothermal power infrastructure could provide even greater efficiency and utilization of resources while possibly improving capacity factors in aged geothermal installations [2]. Geothermal capacity factors in the United States averaged 69.6% in 2019 [3] such that optimized planning and system management could enable opportunities to uplift geothermal power generation and displace carbon dioxide emitting fuel sources.

While readily available, geothermal power generation technology has not changed significantly since the first power captured at Larderello in Italy using direct-steam technology [4] or the pioneering works to use liquid-dominated geothermal reservoirs at Kawerau and Wairakei in New Zealand [5]. Binary technologies using iso-pentane and other similar working fluids have allowed lower-temperature fluids to be used but still rely on infrastructure similar to that used by steam turbine technology [4]. Aged fields often have wells, pipelines, separation units and generators that have been commingled over the life of the geothermal power installation such that brand new turbines may be powered by wells that have operated for decades and vice versa. Operators have done very well to operate these installations using traditional techniques and engineering expertise but to a lesser degree have relied on the application of machine learning (ML), artificial intelligence (AI), and other advanced interrogation technologies to further leverage knowledge and improve operation of these assets. The use of these technologies in geothermal exploration, including the drilling of new wells, is another exciting topic that is currently being explored by several groups in the industry but is not part of this work and is not discussed further in this paper.

Broadly speaking, geothermal operations have not seen the widespread application of digital analytics to guide decision making that other power generation industries have seen. In particular, the use of digital twins has seen widespread adoption in the power industry but not in geothermal. Digital twins that can accurately model a power plant are tremendously useful since they can be manipulated and optimized in a fraction of the time and cost of a real power plant [6,7]. The result is improved efficiency, capacity factors, and overall resource optimization and utilization.

Digital twins are often built using commercial software [8–12] or publicly available software driven by public research [13–16]. Many of these software packages, such as Flownex [10] and RELAP [16], are component-based systems models, which enable users to create digital twins composed of a collection of granular components. These digital components are often simplified models that describe bulk thermodynamic parameters and behaviors using a foundation of theoretical and semi-empirical mass and heat transfer relationships. The simplicity of these models often provides accurate and useful insights while maintaining reasonable computational requirements. Equally important, the component-based granularity makes models easy to interrogate for detailed engineering analysis.

Several challenges have slowed the adoption of systems modeling using data-driven digital twins in the geothermal industry. An incomplete list of these challenges is provided here. First, many of the readily available software packages are focused on non-geothermal systems such as hydrocarbon processing [11], gas turbines [10], data centers [13], and nuclear reactors [16]. Second, systems modeling software commonly requires time intensive and precision tuning of engineering parameters to create accurate models. Many existing geothermal plants are large systems that may have had decades of infrastructure changes, making the collection of precision engineering data in a standard format onerous and impractical. Lastly, many existing software packages such as RELAP [16] are built on decades of theoretical exploration and experimental validations of physical phenomena. Adaptation of these industry-specific software packages to geothermal plants may require significantly expanded research specific to geothermal phenomena. Development of such research can be highly resource intensive and cost-prohibitive.

Overarching these challenges is the fact that traditional systems modeling tools are frequently idealized thermodynamic representations that do not consider the actual performance characteristics of an as-built plant. Modeling assumptions, while acceptable in theory, do not map well to real systems. Thermo-chemico-physical changes to a geothermal system happen the instant a plant is commissioned, and while care and diligence are taken to ensure safety factors are considered, the active performance will drift from design conditions [17]. For example, the work by Hernández et al. [18], which applies the HYSYS modeling software to a simple geothermal system, made the engineering assumption of constant two-phase well mass flow, which may not accurately predict the response in well

deliverability to varying system pressure or the degradation of performance over the well's lifetime. Accordingly, the modeling framework introduced in this paper focuses on hybrid data-driven thermodynamics models alongside historical data assimilation methods to represent the system in a way that accounts for the inaccuracies of modeling assumptions and the imperfections of an as-built system.

In this paper, we present a modeling framework called GOOML (Geothermal Operational Optimization with Machine Learning) for creating digital twins of geothermal power plants based on hybrid data-driven thermodynamics component-based systems models. Instead of relying on extensive theoretical and semi-empirical relationships, we enforce simple first-principal thermodynamic mass and energy conservation equations and then use real historical plant data for training machine learning models to describe the thermodynamic operations of various steamfield components. This method is shown to accurately describe geothermal systems without relying exclusively on theoretical phenomenology and while greatly reducing the required engineering design details. The limitations of this approach are represented in the error metrics presented in Section 3.1 and are further discussed in a cross-validation study in Section 3.2 and the discussion in Section 4. We envisage that the application of data-driven modeling to geothermal systems analysis will provide operators with tools that highlight opportunities within existing infrastructure for optimization. We believe this will help improve the overall system understanding and will likely improve capacity factors and overall system utilization.

## 2. Materials and Methods

### 2.1. Data Sources

GOOML models have been developed for several geothermal systems: the Wairakei Geothermal Field in New Zealand, TOPP1 in New Zealand, and the McGinness Hills Geothermal Complex in the United States. The data and validation results presented in this paper are exclusively from the Wairakei field owing to the fact that this is the most complex GOOML model developed thus far and therefore has the most interesting and extensible results. In total, the Wairakei system model includes 177 individual component models and integrates 3 years of historical data (2018–2020) at a 30-min frequency from 233 data features.

As our research has focused on analyzing active geothermal fields, instrumentation of such installations is not comprehensive, and the resultant data streams are less than fully constrained. Sensor drift and failure, system interruption, and failure to manually record operator actions all contribute to errors and inconsistencies in the data. We performed automated quality assurance and data formatting processes before ingestion of the data into the GOOML model. The data quality assurance, formatting, and ingestion process was recursive in that we regularly requested and received additional data from operators to keep data sets recent, fill gaps, and provide new insights through additional sensor measurements. This is a standard process for preparing real data for use in an analytical space, as gaps and anomalous data points must be backfilled or filtered to ensure that data streams represent real-world events.

### 2.2. System Frameworks

The GOOML modeling ecosystem includes two main system frameworks that organize and execute a networked collection of component objects. The two frameworks are the historical system, which can be used for data assimilation, data cleaning, and regression training; and the forecast system, which can be used along with trained component regression models to predict future system operation.

#### 2.2.1. Historical System

Developing a historical system, or a "historical data model", is the first step taken to build a GOOML model for an operating steamfield. First, a system configuration is developed to organize the general topology of the steamfield network as shown in Figure 1.

Input time-series data are assigned to each component object within the system (e.g., a well gets assigned its respective historical pressure, mass flow, and enthalpy data).

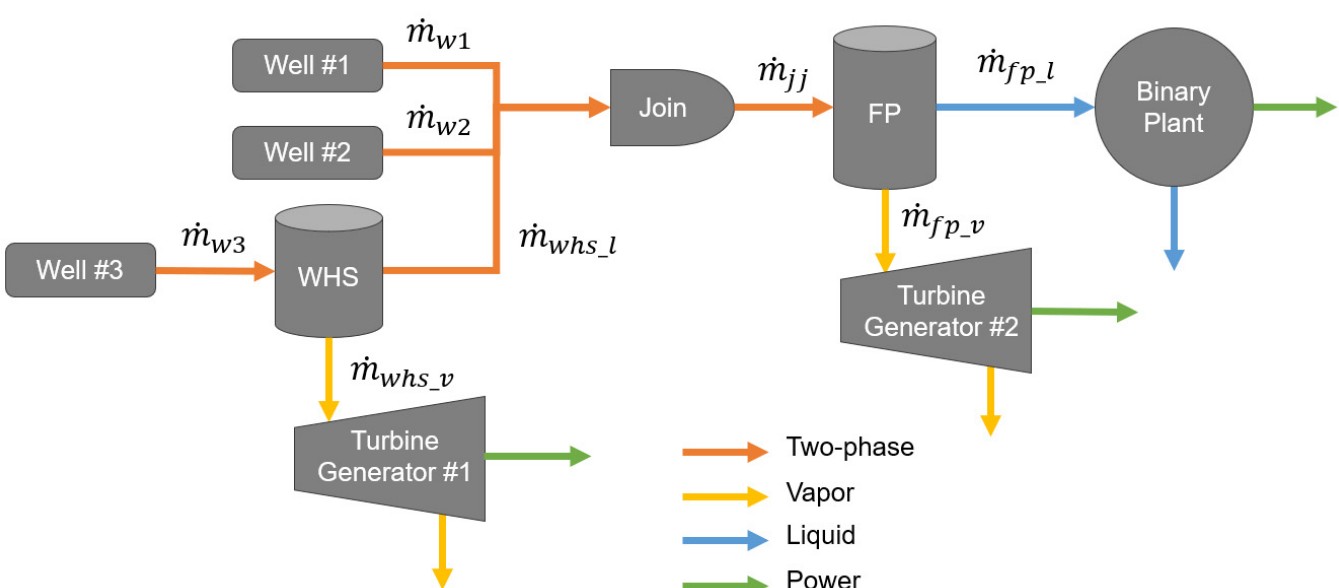

**Figure 1.** Block diagram of a basic GOOML system network where WHS is a well head separator, FP is a central flash plant, $\dot{m}_x$ is the mass flow from component $x$, and additional subscripts $\dot{m}_{x\_v}$ and $\dot{m}_{x\_l}$ represent separated vapor and liquid flow, respectively.

The historical data model attempts to enforce conservation of mass and energy equations by manipulating two-phase flow estimates while preserving trusted and accurate single-phase flow measurements (see Assumption 9). The model also attempts to fill any gaps in the data using thermodynamic relations. One example is the fluid conditions at the inlet of separator units, which is often not instrumented. The pressure at this location is solved for as described in Section 2.3.2, and the enthalpy can then be solved using balance equations.

After cleaning the mass and enthalpy balances and solving for missing thermodynamic system and component properties, the historical system model represents the best estimate of the true historical conditions of the geothermal system. This result by itself is incredibly useful and can be used to interrogate historical plant conditions and to understand how individual components have performed. The object-oriented networked relational structure makes it intuitive to chase down data within an otherwise incredibly complex steamfield. However, beyond simple data interrogation, the historical data model is instrumental in organizing data into a format that can be readily used to train regression models. This provides the foundation for the forecast system.

### 2.2.2. Forecast System

The forecast system is built on the same plant configuration as the historical system, but it trades historical time series data for trained regression models. Instead of a component being defined by its measured historical data, the component performs thermodynamic operations on the geothermal fluid based on its regression model. For example, a turbine forecast model will extract some amount of energy from the working fluid based on the data features of the component's input and possibly some data features from further upstream in the system. The regression model that defines the amount of energy extracted is trained on the performance of the component in the historical model. Conservation of mass and energy is also maintained in the forecast system by constraints on the individual component regression models. Most forecast components can be fully defined by their regression models and the component's input fluid. However, some boundary conditions must be set, such as: deliverability curves of wells, the well head pressures, and turbine inlet pressures. Turbine inlet pressures are assumed to be setpoints controlled by the opera-

tors (Assumption 11), and well deliverability models are further described in Section 2.3.1. With these boundary conditions and component-based regression models, the forecast system is able to efficiently execute simulations that predict possible future operations.

### 2.3. Component Models

The GOOML modeling framework can be thought of as an ecosystem of individual component models that are assembled into arbitrary system frameworks. The basic GOOML components are listed below in Table 1 along with their data inputs and typical model types. The flexibility of the object-oriented GOOML modeling framework also allows for easy experimentation with different forecast model architectures using open-source or custom software.

**Table 1.** Basic GOOML component model descriptions.

| Component | Common Historical Data | Forecast Model | Model Input Features |
|---|---|---|---|
| Single-Phase Well | Pressure, temperature, mass flow | Linear extrapolation with decline | Pressure, temperature, mass flow |
| Two-Phase Well | Pressure, mass flow (TFT estimate), enthalpy (TFT estimate) | TFT deliverability curves with decline | Pressure |
| Separator (Flash Plant) | Output steam pressure, output steam mass flow, output liquid mass flow | Feed-forward neural network, theoretical (thermodynamic-based) | Input mass flow, input enthalpy, input pressure, input steam quality, input velocity, residence time, steam quality at separation pressure, theoretical pressure drop, cyclone design number |
| Turbine Generator | Heat sink temperature, power | Multi-linear regression, Willans Line, theoretical (Carnot-based) | Input mass flow, input enthalpy flow, heat sink temperature, temperature differential |
| Binary Plant | Heat sink temperature, power | Multi-linear regression, theoretical (Carnot-based) | Input enthalpy flow, temperature differential, upstream flow contribution fractions |
| Join Junctions | N/A | N/A | N/A |
| Split Junctions | N/A | N/A | N/A |

### 2.3.1. Well Models

Wells are the conduits of the motive source for any geothermal power plant, and their component models are the starting point of any GOOML model. Single-phase wells can be described with historical pressure, temperature, and mass flow measurements, and their forecast models are simple extrapolations of these values. Single-phase wells can also be assumed at single-phase saturated conditions, making historical temperature (or pressure) an optional input. Two-phase wells are modeled using deliverability curves based on tracer flow tests (TFTs) to estimate their two-phase mass flow and enthalpy as a function of directly measured well head pressure. By basing the well components on TFT equations, the wells can dynamically respond to changes in system operating pressure. The well components allow input TFT equations as either linear or polynomial functions to derive the enthalpy relation and as elliptical relations for mass flow, although the GOOML software can accommodate any number of more exotic functions. Two-phase well forecasts extrapolate their historically observed pressure and use the most recent TFT equations and measurements to estimate future mass flow and enthalpy. All well forecasts can be paired with linear or exponential decline rates based on historical observations.

### 2.3.2. Two-Phase Separator Models ("Flash Plants")

Downstream from two-phase wells are separator units (also called flash plants). These units reduce the pressure of the geothermal fluid by "flashing" saturated liquid into saturated steam and also commonly include an internal cyclone design to further remove suspended water vapor from the steam. This class of components includes well head separators (WHS), which are typically smaller separator units dedicated to a single two-phase well. Separator units are expected to be sparsely instrumented despite their complex thermodynamic operation. Historical mass flow and pressure are required for at least one of the two single-phase outputs. The historical system model solves for separator inlet conditions based on the flow from upstream wells, the turbine inlet setpoint pressure, and the separator pressure drop as shown in Equations (1) and (2) below from Lazalde-Crabtree [19].

$$\Delta P = \frac{NH \times u^2 \rho_v}{2},\tag{1}$$

$$NH = 16\frac{A_i}{D_e^2}\tag{2}$$

where $\Delta P$ is the vapor-phase pressure drop across the separator (commonly assumed to be similar to the liquid-phase pressure drop), $u$ is the superficial vapor-phase inlet velocity, $\rho_v$ is the vapor-phase density, $A_i$ is the inlet cross-sectional area, and $D_e$ is the diameter of the outlet steam pipe.

Separator forecast models are trained to predict the non-dimensional separation efficiency defined by Equation (3) below:

$$eff_{fp} = \dot{m}_v/\dot{m}_{tot}\tag{3}$$

where $eff_{fp}$ is the non-dimensional separation efficiency, $\dot{m}_v$ is the separated vapor-phase output mass flow, and $\dot{m}_{tot}$ is the total component mass flow. The steam qualities of the separated single-phase steam (also sometimes referred to as separation efficiency in the literature) and liquid outputs are assumed to be 1.0 and 0.0, respectively, as per Assumption 7. The forecasted non-dimensional separation efficiency can be predicted by a feed-forward neural network or a simple theoretical calculation of the steam quality at the estimated separation efficiency.

The neural network in this case has three hidden layers, each with 128 nodes modeled in TensorFlow [20]. The layers are fully connected but trained with 50% dropout. All nodes use the rectified linear unit (RELU) activation, which was chosen based on our attempt to linearize some of the training features. We originally performed a full Gaussian hyperparameter search to optimize the model architecture but decided against the "optimized" model architecture because it resulted in saliency maps that were completely unexplainable, non-intuitive, and likely non-physical. This "final" architecture of $3 \times 128$ with 50% dropout resulted in a similar validation error compared to the "optimized" model but also produced explainable and highly intuitive saliency maps that we deemed to be more likely representations of the physical phenomena. The neural network predicted separation efficiency is shown to produce reasonably accurate system-wide results in Section 3 and is shown to be more accurate than the simple theoretical estimate of steam quality at separation pressure.

### 2.3.3. Power Generating Models

Turbine generator units and binary plants are modeled as simple thermodynamic operators that extract enthalpy from the working fluid based on their historical power generation. Typically, only the measured power generation and heat sink temperature are required for historical input, although additional inputs for isentropic efficiency, output steam quality, and output pressure are also accepted. The turbine generator and binary plant forecast objects predict the power generation based on the thermodynamic conditions of the input streams. Complex models such as neural networks were tested for these

components but were disregarded owing to the minimal available data inputs. Instead, the power generation is typically modeled using a multi-linear regression, which has shown significant improvement over a simple mass-to-power ratio (as in the Willans Line [21]). The turbine generator models also include inputs for a no-load steam requirement and a maximum power generation value. There is also an option to model the power generating units using an idealized Carnot efficiency with multiplicative and additive adjustment terms, which can be useful if historical data are unavailable or inaccurate.

### 2.3.4. General Utility Component Models

There are also several generic components that are used in both the historical and forecast systems, sometimes without data inputs to combine and separate flows. These components are referred to as join junctions and split junctions. Join junctions combine multiple input flows into a single output. This is a common operation to collect two-phase mass flow from multiple wells to deliver to a large, centralized separator unit. The join junctions are governed by the following balance equations:

$$\dot{m}_{out} = \sum_{i=1}^{n} \dot{m}_i,\tag{4}$$

$$h_{out} = \frac{\sum_{i=1}^{n} h_i \times \dot{m}_i}{\dot{m}_{out}}\tag{5}$$

where $\dot{m}_{out}$ and $h_{out}$ are the junction output mass and specific enthalpies, $n$ is the total number of input flows being joined, and $\dot{m}_i$ and $h_i$ are the input mass flow and specific enthalpy of input flow $i$. These equations conserve mass and energy across the junction. The join junction output pressure is either assumed to be equal to the minimum input pressure (Assumption 10) or is solved for by downstream components (e.g., by a separator pressure drop equation).

Split junctions divide flow into two outputs. The split junction provides the basic template for the two-phase separator component. These can also be used to model steam delivery to industrial users or to vent steam when operational limits dictate. Split junctions are governed by the same balance equations as the join junctions:

$$\dot{m}_{in} = \dot{m}_{out1} + \dot{m}_{out2},\tag{6}$$

$$\dot{m}_{in}h_{in} = \dot{m}_{out1}h_{out1} + \dot{m}_{out2}h_{out2}\tag{7}$$

where the mass into the split junction $\dot{m}_{in}$ must be equivalent to the sum of the split output mass flows and same as the enthalpy flow $\dot{m}_{in}h_{in}$. In order for the split junction equations to be solved, certain assumptions must be made about the distribution of the mass and enthalpy to the outputs. For example, with two-phase separators, the division of output mass flow is either measured or predicted by $eff_{fp}$ and the enthalpy of the output flows is determined as a function of the output pressures and steam qualities. Other types of split junction such as a vent component distribute all input mass flow over a certain operational threshold to the venting output, while the other output receives the mass flow at or below the threshold. In this case, the specific enthalpies of the two outputs are assumed to be equal to the input since no separation is performed.

While the component models above are intended to be simple, they form the basis upon which highly complex systems can be built. New component models can be easily created by inheriting these pre-existing templates or being created from something completely new. When networked together in a system model, the components operate on the thermodynamic working fluid and can create valuable insights and predictions.

### 2.4. Modeling Assumptions

All engineering models make assumptions about the physical systems they represent. The GOOML modeling ecosystem is no different and understanding these assumptions is

critical to proper application of the software. In fact, because of the novelty of the GOOML modeling approach, there are perhaps even more assumptions than are typical in this type of work. Nevertheless, the power of the GOOML approach lies in the use of real data to compensate for the simplifications made in the assumptions. The assumptions are enumerated below with descriptions and justifications. Readers should consider the overarching philosophy behind these assumptions: simplification of systems to the minimum complexity that maintains usefulness, reduction of computational complexity to allow investigation in more rigorous and resource-intensive ML and AI experiments, and reliance on real historical data as much as possible instead of prescribed physical relationships. In this spirit, we made the following assumptions:

**Assumption 1.** *The geothermal working fluid is assumed to have the thermophysical properties of pure water, with data used from the NIST Chemistry WebBook [22]. Thermophysical properties specifically tuned to site-specific geothermal brines may be more accurate on a case-by-case basis but using the readily available pure water properties is intended to be more broadly applicable for a generic modeling tool. As with many of the subsequent assumptions, training regression models on real historical data is expected to compensate for the inaccuracies introduced by this assumption.*

**Assumption 2.** *Two-phase geothermal fluid is assumed to be in thermodynamic equilibrium. That is, no inter-phase heat transfer is modeled. This is a physically reasonable simplification, given the geothermal source of the working fluid and the absence of detailed heating/cooling models.*

**Assumption 3.** *Thermodynamic processes are assumed to be adiabatic. That is, parasitic losses to the environment are not modeled explicitly. This is a necessary assumption, given the lack of measurements quantifying the parasitic losses. This assumption is at least partially handled by the efficiencies predicted by the data-driven models and could be further refined in the future by adding environmental loss terms.*

**Assumption 4.** *A steady-state approximation is assumed. Dynamic systems are analyzed from timestep-to-timestep. This is an appropriate assumption for analyzing long term geothermal forecasts on the scale of days to years but would not be appropriate for analyzing short-term events such as a pipe burst or other system failures.*

**Assumption 5.** *Mass and energy balances are enforced; the momentum balance is not. This is an appropriate assumption, given the coarse component-based resolution of the modeling system.*

**Assumption 6.** *Pressure losses in system pipes are not modeled explicitly. This is an acceptable assumption because piping losses were estimated to be minor compared to the driving pressures. These losses are also at least partially handled by the data-driven models, which are trained on data that aggregate these losses. Pressure loss component models can be easily added in the future.*

**Assumption 7.** *The output of two-phase separator units (also called "flash plants") are assumed to be ideal saturated steam and liquid with steam quality of 1.0 and 0.0, respectively. Examples in the literature noted that separated steam quality from a cyclonic separator is typically greater than 0.997, but since this is not measured directly in the power plant data we used, it is assumed to be 1.0 [19,23,24].*

**Assumption 8.** *If not measured explicitly, the liquid output pressure of a two-phase separator is assumed to be equal to the steam output pressure, which is often measured directly in generation processes. This is an appropriate assumption, given the lack of instrumentation on such installations, and has been shown to provide accurate results when modeling the separated geothermal liquid such as in a binary plant system.*

**Assumption 9.** *When a mass balance in the historical data model is over-constrained (e.g., if all three mass flow terms were supplied to Equation (6)), single-phase flow measurements are assumed to be accurate, and two-phase flow estimates (often based on two-phase well TFT equations) are corrected using the ratio of the initial mass flow estimate to the over-constrained measurements. The upstream enthalpy is also corrected using the over-constrained enthalpy balance. Over-constrained balance equations with only single-phase flow measurements raise an error and the user is required*

*to remove data streams from the configuration until the mass flow can be balanced without manipulating single-phase flow measurements. By removing the single-phase flow measurements from the configuration, the user can also choose to preserve the two-phase flow estimates. This assumption is invaluable for creating a physically consistent data set that conserves mass and energy.*

**Assumption 10.** *When not measured directly, the pressure at the output of a join junction is assumed to be the minimum of the input pressures. If a join junction is the input to a flash plant with a measured output pressure, the pressure of the join junction is iteratively calculated using (1). The thermodynamic state at the output of the join junction is then calculated as a function of the mixture enthalpy (calculated using Equation (5)) and the pressure. This assumption helps reduce computational complexity and input requirements to the model.*

**Assumption 11.** *Several components in GOOML forecast systems are assumed to drive the solution for pressure conditions. For example, forecasted well objects are assumed to be discharging at the nominal discharge pressure observed in the historical data. Turbine inlet pressures (and therefore also flash plant output pressures) are assumed to be fixed values controlled by the operators. Intermediate pressures are solved for when pressure drop equations are available and can be solved (such as the inlet pressure at a flash plant iteratively solved using Equation (1)). This assumption is based on discussions with geothermal operators and has been shown to provide a fair approximation of the system.*

**Assumption 12.** *Power generating components (turbine generators and binary plants) are often assumed to extract enthalpy from the working fluid with perfect isentropic efficiency. The pressure drop across power-generating components is assumed to be zero such that the output state can be defined using input pressure and output enthalpy. While these are non-traditional assumptions, we believe they are acceptable given that we target the prediction of turbine power generation and not conditions at the outlet of the turbine. Alternative assumptions commonly result in non-physical thermodynamic states, although these assumptions can be easily revised by user input when fully constrained data informs model development.*

## 3. Results

The results presented here compared the GOOML historical Wairakei model to the GOOML forecast model for the 2018–2020 operating period. The historical model collated historical data, such that the data presented by this model were directly measured from components in the steamfield during operations. However, some system-level metrics were aggregated from several measurement sources, such as the total system mass take or the total steam line flows to various power stations. In contrast, the forecast model primarily represented predicted system and component level metrics. It is important to note that the forecast model was "seeded" with historical well and turbine pressures, heat sink temperatures, and operator actions in order to provide a fair evaluation of the predictive capabilities of the regression models (it was not within the scope of this research to forecast ambient heat sink temperatures or operator actions).

Forecast models for production wells were trained on an abbreviated period of stable operation during June and July of 2018. Several wells had work-overs that resulted in increased mass flow and changes to enthalpy performed in the 2018–2020 timeframe that were re-baselined in the forecast model to reflect these changes. Other data-driven regression models (flash plants, turbine generators, and binary plants) were trained using 30-min data from 2018 and 2019 (approximately 35,000 data records). A single neural network was trained to model all flash plants while separate models were trained for each individual turbine generator and binary plant. This granularity in models was decided upon because of the engineering details that were available to describe and differentiate the flash plants, but not the turbine generators or binary plants.

*3.1. Forecast Model Validation*

　　　The results of 3 years of historical data contrasted with a nominal forecast model for the Wairakei Geothermal Field are presented in Figures 2–9 and Table 2 below. Note that the Wairakei Geothermal Field includes three separate power stations: Wairakei, Te Mihi, and Poihipi. Alongside the nominal forecast model are two simplified models with a focus on theoretical versus data-driven models. The "Theoretical FP" model predicts the flash plant separation efficiency to be equivalent to the steam quality at the estimated separation pressure. The "Willans Line TG" model predicts the turbine generator power output using a constant steam consumption rate (the "Willans Line" approximation) provided by the power plant engineers.

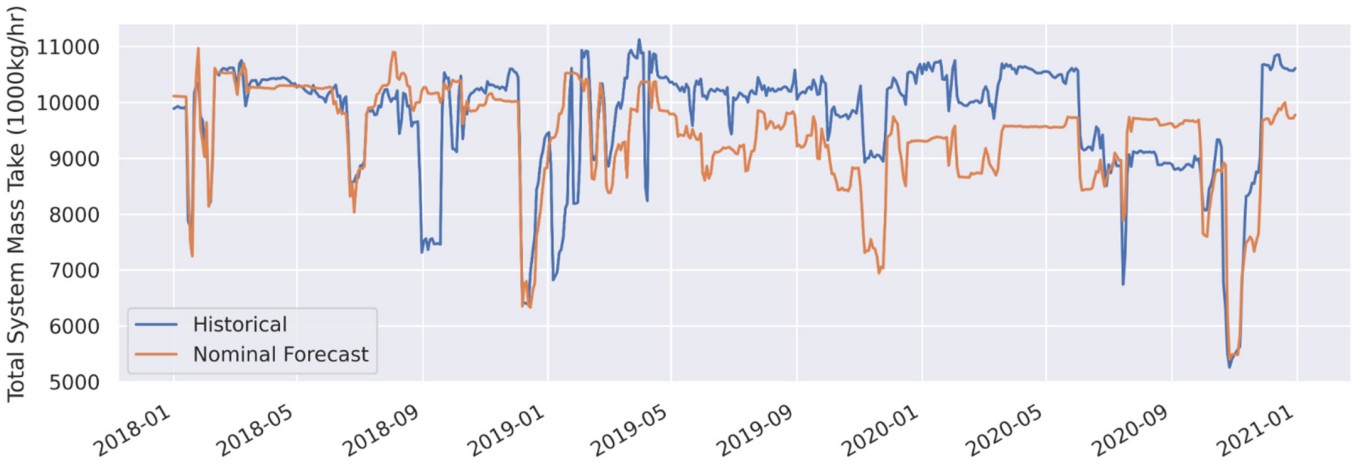

**Figure 2.** Total system mass take, comparing historically measured data versus the nominal forecast model.

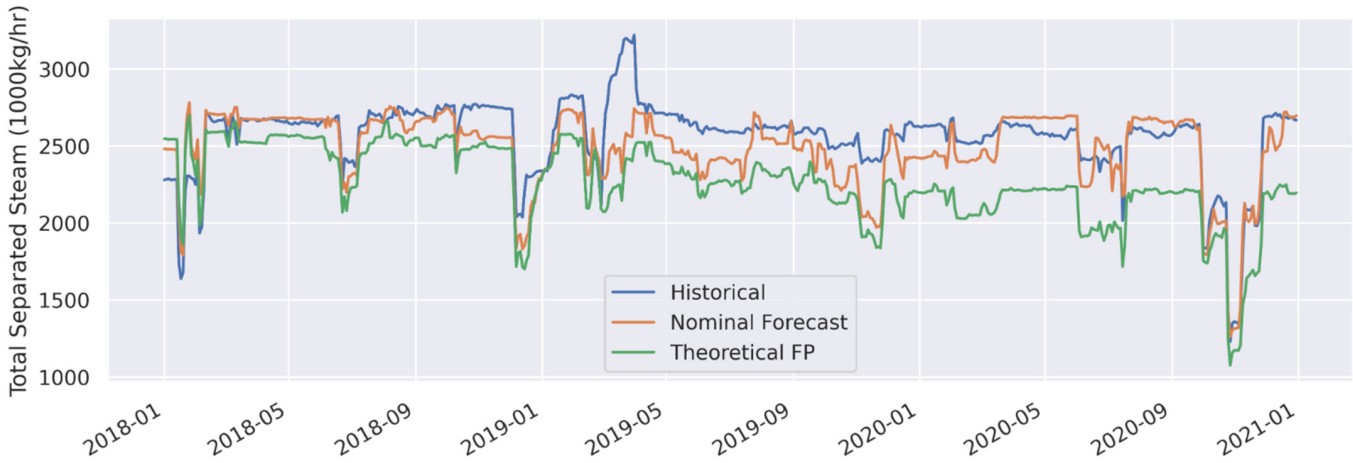

**Figure 3.** Total system separated steam flow, comparing historically measured data versus the nominal forecast model (including the neural network flash plant model) and the theoretical flash plant forecast (using steam quality at separation pressure as separation efficiency).

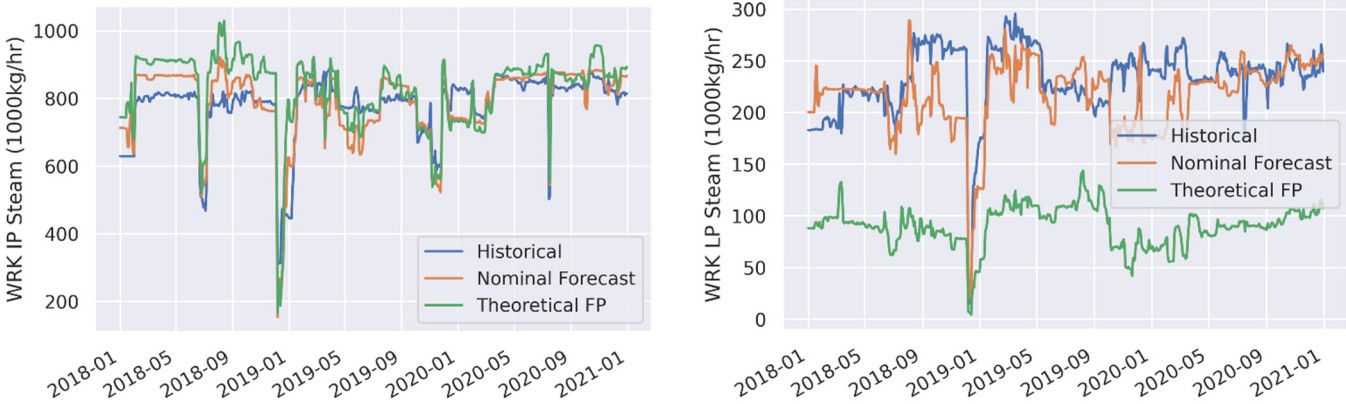

**Figure 4.** Intermediate pressure (**left**) and low pressure (**right**) separated steam flow to the Wairakei power station, comparing historically measured data versus the nominal forecast model (including the neural network flash plant model) and the theoretical flash plant forecast (using steam quality at separation pressure as separation efficiency).

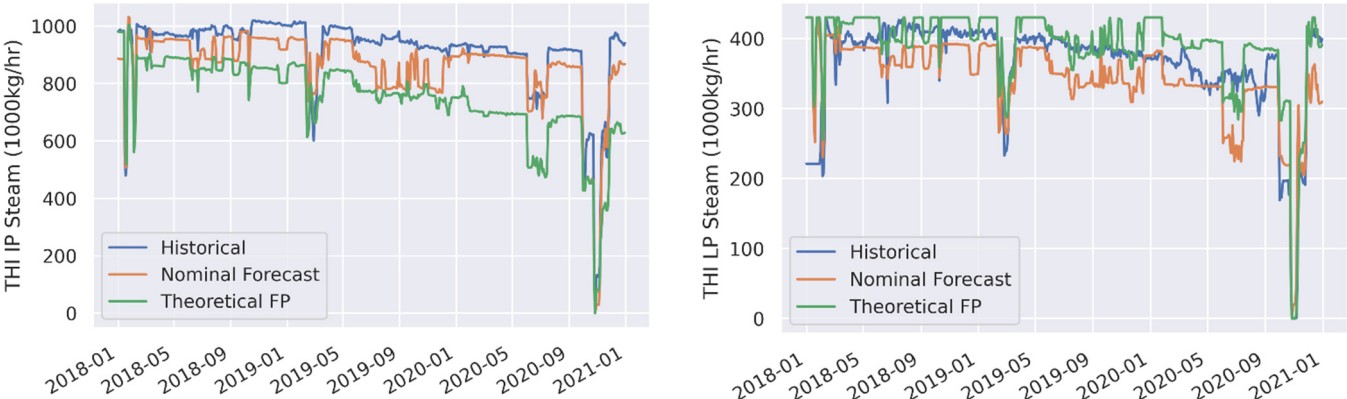

**Figure 5.** Intermediate pressure (**left**) and low pressure (**right**) separated steam flow to the Te Mihi power station, comparing historically measured data versus the nominal forecast model (including the neural network flash plant model) and the theoretical flash plant forecast (using steam quality at separation pressure as separation efficiency).

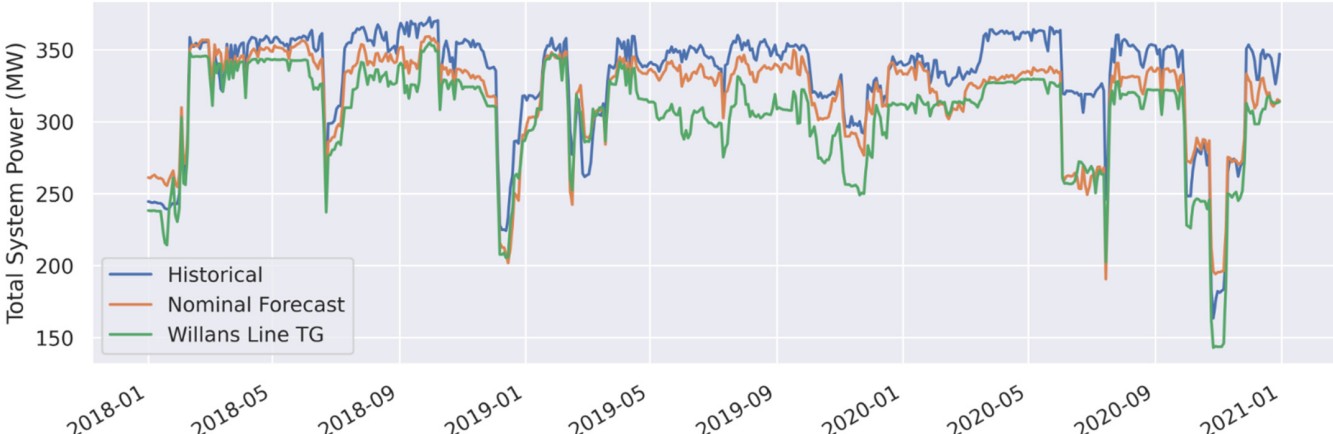

**Figure 6.** Total system power generation, comparing historically measured data versus the nominal forecast model (including the turbine generator multi-linear regression model) and the Willans Line turbine generator forecast (using a constant value for turbine generator steam consumption rate).

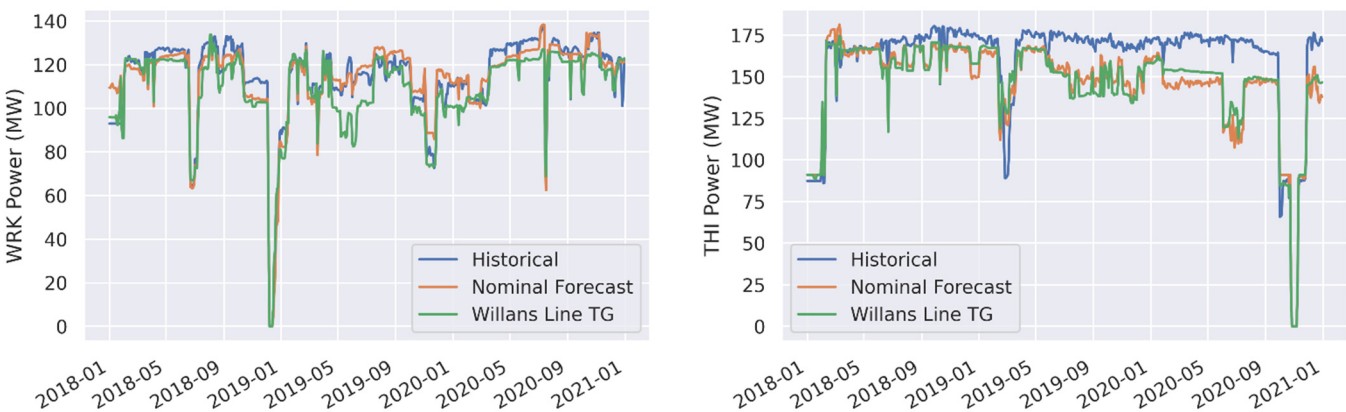

**Figure 7.** Wairakei (**left**) and Te Mihi (**right**) system power generation, comparing historically measured data versus the nominal forecast model (including the turbine generator multi-linear regression model) and the Willans Line turbine generator forecast (using a constant value for turbine generator steam consumption rate).

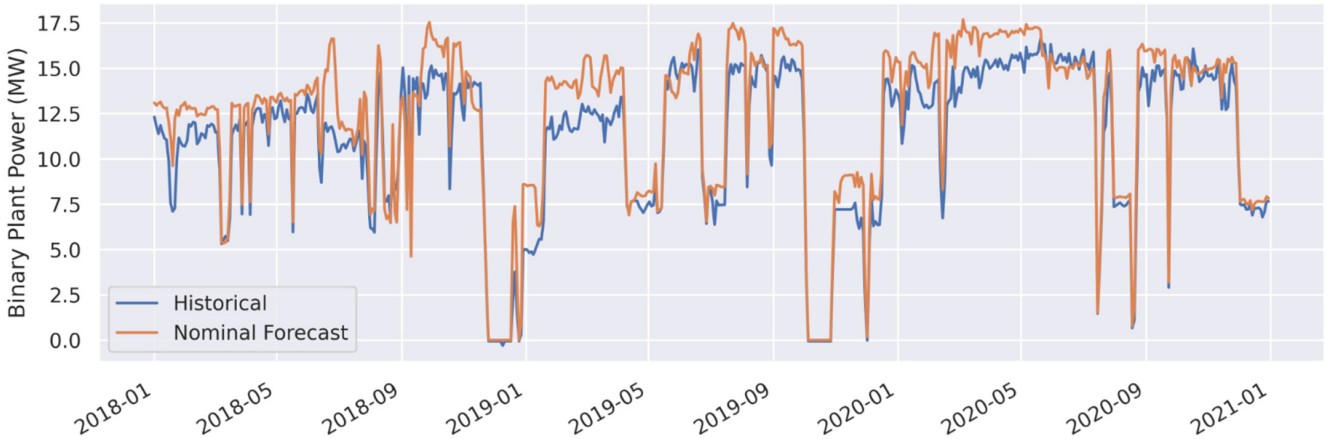

**Figure 8.** Binary plant power generation, comparing historically measured data versus the nominal forecast model (including the binary plant multi-linear regression model).

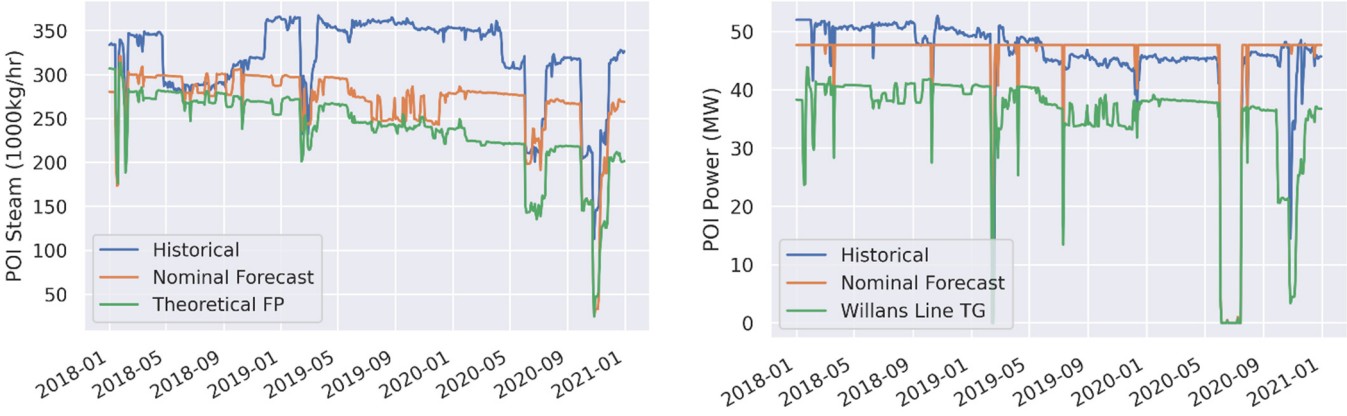

**Figure 9.** Poihipi Road separated steam flow (**left**) and turbine generator power (**right**), comparing the historically measured data versus the nominal forecast model and the theoretical flash plant and turbine generator models.

**Table 2.** Comparison of results of steamfield model components in the GOOML environment based on mean bias error (MBE) and mean absolute error (MAE) between historical and forecast data. Relative error values are calculated with respect to the mean historical value.

| Metric | Model | MAE | MAE (%) | MBE | MBE (%) |
|---|---|---|---|---|---|
| Total System Mass Take (1000 kg/h) | Nominal Forecast | 783 | 8.1 | −328.4 | −3.4 |
| Total Separated Steam (1000 kg/h) | Nominal Forecast | 133 | 5.2 | −76.8 | −3.0 |
| Total Separated Steam (1000 kg/h) | Theoretical FP | 314.2 | 12.3 | −297 | −11.6 |
| WRK IP Steam (1000 kg/h) | Nominal Forecast | 47.5 | 6.1 | 10.9 | 1.4 |
| WRK IP Steam (1000 kg/h) | Theoretical FP | 70.7 | 9.1 | 45.4 | 5.8 |
| WRK LP Steam (1000 kg/h) | Nominal Forecast | 29.3 | 12.6 | −16 | −6.9 |
| WRK LP Steam (1000 kg/h) | Theoretical FP | 142.8 | 61.3 | −142.8 | −61.3 |
| THI IP Steam (1000 kg/h) | Nominal Forecast | 62.5 | 6.8 | −54.2 | −5.9 |
| THI IP Steam (1000 kg/h) | Theoretical FP | 166.6 | 18.2 | −163.8 | −17.9 |
| THI LP Steam (1000 kg/h) | Nominal Forecast | 38.3 | 10.5 | −20.2 | −5.5 |
| THI LP Steam (1000 kg/h) | Theoretical FP | 34.6 | 9.5 | 25.2 | 6.9 |
| POI Steam (1000 kg/h) | Nominal Forecast | 52.3 | 16.4 | −49.8 | −15.6 |
| POI Steam (1000 kg/h) | Theoretical FP | 81.1 | 25.4 | −80.7 | −25.3 |
| Total System Power (Gross MWe) | Nominal Forecast | 16.6 | 5.0 | −12.8 | −3.9 |
| Total System Power (Gross MWe) | Willans Line TG | 28.5 | 8.6 | −27.4 | −8.3 |
| WRK Power (Gross MWe) | Nominal Forecast | 4.8 | 4.2 | −0.3 | −0.3 |
| WRK Power (Gross MWe) | Willans Line TG | 6.8 | 5.9 | −5.4 | −4.7 |
| THI Power (Gross MWe) | Nominal Forecast | 16.7 | 10.4 | −14 | −8.7 |
| THI Power (Gross MWe) | Willans Line TG | 15.9 | 9.9 | −13.5 | −8.4 |
| POI Power (Gross MWe) | Nominal Forecast | 2.7 | 6.1 | 0.4 | 1.0 |
| POI Power (Gross MWe) | Willans Line TG | 9.8 | 21.9 | −9.6 | −21.5 |
| Binary Plant Power (Gross MWe) | Nominal Forecast | 1.3 | 11.8 | 1.1 | 10.0 |

The total system mass take presented in Figure 2 is the sum of two-phase mass flow from all wells in the system. Overall, the forecast showed fairly accurate results compared to the historical mass take, with production well interruptions and maintenance shut events predicted accurately. As mentioned previously, forecast models of wells were trained on an abbreviated period of stable operation during June and July of 2018; deviation in the forecasted mass take that strayed from the training period is unsurprising, given the comparatively long-term forecast.

The forecasted separated steam mass flow delivered to the turbines presented in Figures 3–5 was accurately predicted by the flash plant machine learning models. In nearly every case, the machine learning model was more accurate than the simplified theoretical thermodynamics-only model. An exceptional improvement is seen in the Wairakei low pressure steam in Figure 4, where the machine learning model and the simple theoretical model had relative mean absolute errors of 12.6 and 61.3%, respectively. The simple theoretical model performed adequately for some components but could not come close to capturing the realities of the as-built system in others. It was reassuring, however, to see that the machine learning model predicted time series features and patterns similar to those of the simple theoretical model; the thermodynamics of the system should create the same general trends in both.

The turbine generator power outputs in Figures 6 and 7 and binary plant power outputs in Figure 8 were accurately predicted by the multi-linear regression forecast models. The trained regression models were shown to predict the historical power within a relative mean absolute error of 5.0%, while the simple Willans Line models had a higher

error of 8.6%. Some of the turbine generators, such as those at the Wairakei power station, are shown to be predicted with exceptional accuracy in Figure 7. Most of the error in the power forecast can be attributed to consistent bias in predicted Te Mihi power, which in turn can be attributed to consistent bias in the predicted Te Mihi steam delivery. One feature of the GOOML modeling framework is the easy ability to add user-specified scalar and adder terms to bias-correct the forecast models. Although not used for these results in order to present a fair evaluation of the trained models, it is likely that the error in the Te Mihi steam and power could be reduced with a simple bias correction term.

It is important to note that the data streams feeding the flash plants and turbine generator regression models in the forecast system were fed data from well forecasts, which were not equivalent to the well data from the historical training set. Furthermore, no degradation in machine learning model performance was seen in 2020 despite the fact that the models were trained using data from 2018 and 2019. These validation results supported the ability of the models to generalize to new operational data.

As with all models, it is important to present, discuss, and understand negative results. In the figures below, several examples are presented where the GOOML forecast does a poor job at matching historical events. For example, the temporary decrease in total system mass take of nearly 2500 tonnes/hour around 2018–2019 is not represented in the forecast model at all as shown in Figure 2. It is quite possible that this data feature was actually an artifact of an out-of-service liquid mass flow sensor, based on the fact that the measured separated steam flow in Figure 3 shows no simultaneous reduction in mass flow. These data artifacts are common in a system as complex as Wairakei and are difficult to resolve completely. Another example is the bifurcation of historical and forecasted mass take after 2019-5 without the related bifurcation in separated steam flow. As noted above, the well forecast models were baselined in 2018, so this long-term deviation is not necessarily surprising. However, it is important to also consider the great work that the plant operators do in keeping the system operating at maximum capacity. Many operator actions are not yet explicitly represented in the GOOML system, and so it is likely this disconnect led to the presented discrepancy. Finally, a number of odd behaviors can be seen in the prediction of the Poihipi power station in Figure 9. This result is an example of the limitations of the GOOML framework coupled with poorly constrained data inputs. In this case, the measured steam flow, pressure, and power at the Poihipi power station resulted in negative mass–enthalpy–power relationships that were clearly non-physical (less steam and lower enthalpy should not result in more power). As such, the weight terms in the multi-linear regression for all turbine generators were bounded with a minimum value of zero, resulting in a constant forecasted power output at Poihipi. Improved data could provide a more realistic forecast, but we have not been able to determine the root cause of these non-physical correlations.

### 3.2. Cross-Validation and Extensibility

When presenting results from a machine learning model, it is important to understand how the model performs with data features outside of its training experience. The multi-linear regressions used for turbine generators and binary plants included threshold values for no-load steam and maximum power generation and so were somewhat protected against significant deviations and unexpected predictions. The flash plant neural network models, however, represented excellent case studies in how machine learning models extrapolate predictions of physics-based component operation. To study this, we trained several models: the first was the nominal forecast model which was trained on all available historical flash plant data from 2018 and 2019. The second was the simple theoretical thermodynamics-only flash plant model (same model that was presented in Section 3.1). The last was a cross-validation forecast ("XVal Forecast") that was trained on flash plant data only when the two-phase input flow was less than 1900 tons/hr. All data observations that exceeded this limit were thrown out of the training set. This threshold was chosen specifically to remove the upper portion of observations for FP16IP+, a large, centralized

separator in the Te Mihi side of the field. Results from FP16IP+ are presented in Figure 10 that show the model's extrapolation behavior when the input mass flow exceeded the cross-validation threshold.

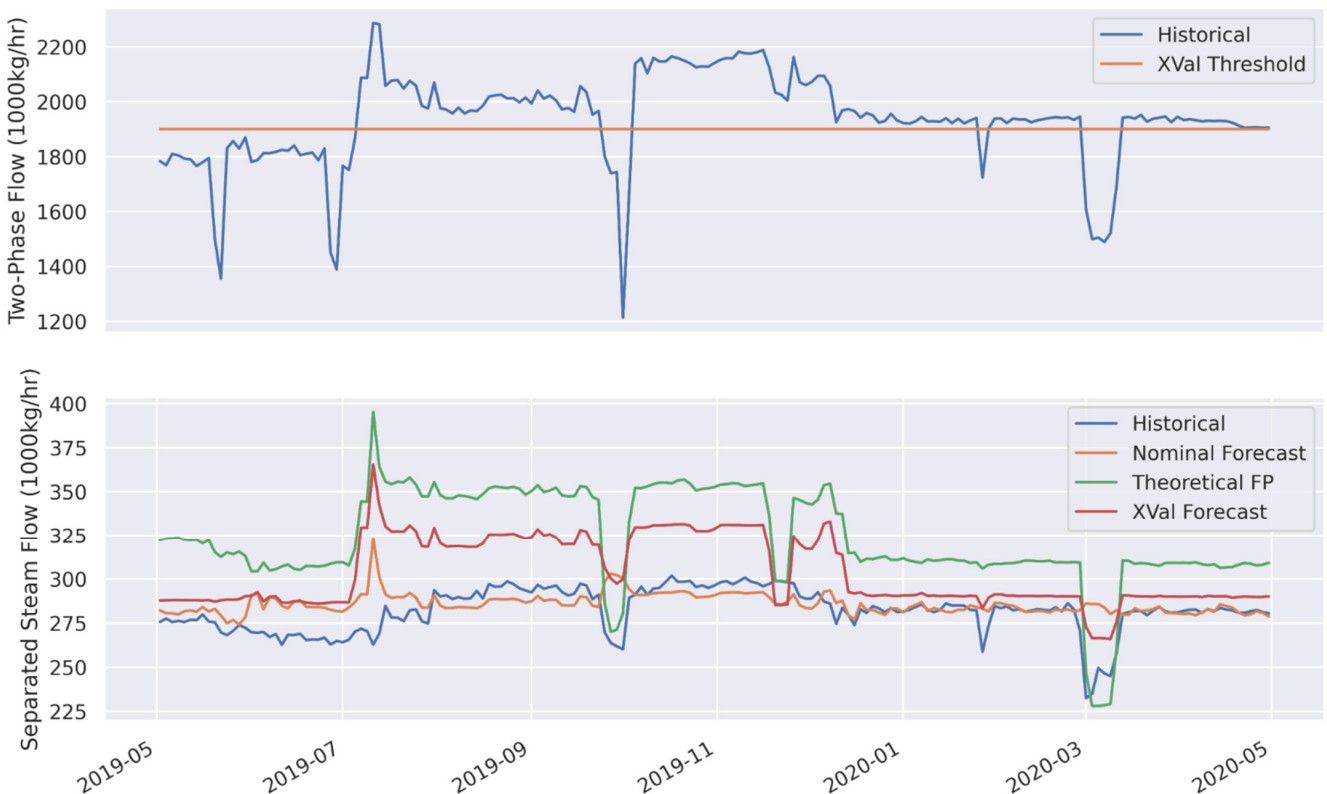

**Figure 10.** Results from a flash plant modeling cross-validation experiment where models were trained only with two-phase input mass flow less than 1900 tonne/h. The top plot shows the two-phase mass flow through FP16IP+ with the training data limitation. The bottom plot shows the historical separated steam flow compared to predictions by a baseline model trained on all data, the cross-validation (xval) model, and a theoretical model.

Examining the forecasted separated steam flow predictions in Figure 10, the cross-validation model did not perform as well as the nominal forecast model when the two-phase mass flow exceeded the cross-validation threshold. However, the cross-validation forecast was able to follow the general trends of the theoretical model even under conditions that it had not experienced. This was likely due to the flash plant neural network model being trained with the steam quality at separation pressure as a training feature. Despite not having been exposed to the high mass flow conditions, the model appeared to linearly extrapolate its predictions, guided by the theoretical performance data feature in the absence of explicit experience. This observation was consistent with previous studies of neural network extrapolation by Xu et al. [25]. This was an encouraging result because it indicated that the machine learning models would not completely fall apart and predict nonsensical data under new observations. Instead, the models appeared to learn the anticipated physical relationships.

Typically, an increase in total input mass flow without a significant change in other data features would cause an increase in separated steam, as predicted by the forecast models just after 2019-07. However, this was not observed in the historical data. In fact, just the opposite was observed, with a small decrease in separated steam. It is possible that this was another data artifact, where the operators may have vented steam directly from the flash plant, reducing the actual observed separated steam flow. It is also possible that this was a sensor anomaly recorded by the liquid flow measurement, which registered an anomalously large total mass flow without a change in the measured steam flow. In either

case, the increase in separated steam mass flow predicted by the forecast models during this event was reasonable based on the physical parameters of the event.

In contrast to the momentary increase in mass flow, the decrease in mass flow around 2019-10 and just after 2020-03 was well predicted by the theoretical model and the cross-validation model, but not the nominal forecast model. During this event, the input steam quality and two-phase specific enthalpy increased significantly. This apparently led the nominal forecast model to predict a commensurate increase in separation efficiency to the extent that the decrease in total mass flow would not be observable in the separated steam flow. Although this prediction was not observed in the historical data or the theoretical forecast, it seemed to be a logical prediction based on the steam quality and enthalpy trends.

Overall, the results of this cross-validation experiment are quite encouraging. Despite the inaccurate predictions under cross-validation conditions, the model was able to follow the theoretical trends when lacking valid training experience. This is exactly the behavior we would like to see in the models and is the primary reason for including the steam quality at separation pressure as a training feature. Extending data-driven models past their training basis is always a risky endeavor, and GOOML users should do so with caution, but these results showed that the flash plant models may be able to do so at least to a limited extent.

## 4. Discussion

To the best of our knowledge, this is the first component-based systems modeling framework with integrated machine learning that was applied to geothermal operations. Similar models have been introduced for industrial chemical processes [26] and building energy systems [27], but not yet for geothermal processes. The work most similar to GOOML is the deep learning modeling methodology introduced by Liu et al. [28], which still exhibits several key differences: GOOML places great emphasis on the conservation of mass and energy within a component-based systems model, while the work by Liu et al. [28] took a data-first approach without considerations for first-principle thermodynamics or system topology. This makes the work done by Liu et al. [28] highly applicable to fault detection tasks but limited to the boundaries of the observed system and training data. The GOOML framework excels at modeling highly complex systems like the Wairakei steamfield with thermodynamically-consistent data features at the input and output of every component. Thanks to the flexibility of the GOOML component-based systems modeling framework, an analyst can easily add or remove components from a model and still get useful performance predictions from the new system topography, allowing them to explore the potential impact of steamfield expansion or component retrofits. Because each component provides an individual contribution to the system (as in real life), an analyst can disable a single component that has never been offline in the training data and the system would respond appropriately. This is demonstrated in the Te Mihi turbine outages around 2020-11 in Figures 6 and 7, which are events that were not present in the 2018–2019 training data but were accurately predicted.

The GOOML model validation results in Section 3 are encouraging and should be sufficiently accurate to provide useful insights for geothermal operations teams. While some model predictions have room for improvement (e.g., Poihipi Road generator), most are highly accurate (the Wairakei power generation), and others can be easily bias-corrected when applied to real-world operations (e.g., the Te Mihi power generation). The accuracy in cross-validation experiments (Section 3.2) was reduced from the nominal validation results, but still followed logical thermodynamic trends and should be useful to provide high-level system insights.

The ability to rapidly retrieve system parameters such as the mass flow or enthalpy for any component in the system is a valuable capability that is currently missing from geothermal operations. Without such a capability, interrogating the system in a root-cause analysis is terribly onerous and inaccurate.

Finally, it is important to consider the computational requirements of system models and the potential for downstream analysis and optimization. The Wairakei historical and forecast models presented here were executed on a standard laptop with a 1.90 GHz processor and 16 GB of RAM. Each model ran in less than 5 minutes, even with the extended 3-year time series at a 30-min data frequency. A single time step can be computed in a fraction of a second, enabling many downstream applications of this modeling framework. For example, some initial research has demonstrated the possibility for a reinforcement learning agent to learn from a GOOML system model. The agent interacts with the system model; it is able to throttle wells, redirect steam, and even perform required maintenance. Because of the computational efficiency of a GOOML simulation, the agent is able to perform many actions per second, learning rapidly and exploring the operational space in a completely new way. Previous literature has described similar methods for optimizing the cooling of large data centers with significant improvements in efficiency [29,30]. We hope to soon demonstrate similar improvements driven by the GOOML modeling framework at real steamfields around the world.

## 5. Conclusions

We set out to build a framework that satisfied the need of the geothermal industry for a simple, yet powerful modeling tool tailored to the unique traits of geothermal steamfields. We believe that GOOML can be used globally by geothermal developers to develop digital twins that will help maximize outputs, optimize processes, and sustainably manage their operations. This will then help increase the competitiveness and uptake of geothermal energy generation worldwide through greater cost effectiveness.

Many assumptions were made in the development of the GOOML modeling framework, made necessary because of incomplete plant data. This process identified that there is a case for improved standardization of geothermal operations data and additional instrumentation in existing and new geothermal fields to better leverage accurate thermodynamic data sets to inform future operations, optimization, and other machine learning efforts. Future work on the GOOML software may focus on refining these assumptions and tools.

While the goal of this paper is limited to presenting the GOOML framework as a predictive tool, future work will demonstrate ways in which GOOML enables advanced operational decision making with true state-of-the-art optimization of real-world geothermal operations. This will include the integration of advanced AI and ML approaches, providing the geothermal sector with additional digital tools to glean new system insights and challenge existing bias and status quo thinking. We hope that this advances the utilization of machine learning in geothermal operations, and we welcome the continued proliferation of such techniques at steamfields around the world.

**Author Contributions:** G.B. was the mastermind behind the GOOML model and lead writer. P.S. was a subject matter expert, second author, and project coordinator. N.T. cleaned, curated, and organized copious geothermal data sets and assisted with ML experiments. M.R. is a neural network beast. J.W. was a key thought leader, data scientist, and project manager. A.B. ensured that real operational data were accessible. J.H. coordinated massive international data transfers. C.S. guided steamfield architecture development. A.U. gave insight into and assured accuracy of data locations. W.M. provided access to data sets and expertise. J.C. measured components in the field. J.Q. provided well-side expertise and steamfield knowledge. R.W. made resources and data sets available. J.A. gave data sets and subject matter knowledge. All authors have read and agreed to the published version of the manuscript.

**Funding:** This material is based upon work supported by the U.S. Department of Energy's Office of Energy Efficiency and Renewable Energy (EERE) under the Geothermal Technology Office Award Number DE-EE0008766. This work was authored by Upflow, Ltd. and the National Renewable Energy Laboratory, operated by the Alliance for Sustainable Energy, LLC, for the U.S. Department of Energy (DOE) under Contract No. DE-AC36-08GO28308 with funding provided by the U.S. Department of Energy Office of Energy Efficiency and Renewable Energy (EERE) Geothermal Technologies Office (GTO). The views expressed in the article do not necessarily represent the views of the DOE or the

**Data Availability Statement:** These works were developed based on data sets that are proprietary in nature. Because of the ownership of these data sets, not all information may be publicly disclosed at the time of publication. The authors endeavor that the information contained herein is based on fully reproducible methods and approaches that are agnostic of the original proprietary data used in development of the work we delivered. Fictional examples of plant configurations and input time series data can be found at the Geothermal Data Repository [31].

**Acknowledgments:** The authors of this paper would like to express our sincere gratitude to our commercial partners: Contact Energy, Ngati Tuwharetoa Geothermal Assets, and Ormat Technologies, Inc., for their support of this project. Without access to their data, expertise and institutional knowledge, this project would not have been feasible. The U.S. Department of Energy's Office of Energy Efficiency and Renewable Energy (EERE) Geothermal Technology Office has graciously supported this project from a concept to reality, and we gratefully acknowledge their enthusiasm for this program of work.

**Conflicts of Interest:** The authors declare no conflict of interests. The funders had no role in the design of the study; in the collection, analyses, or interpretation of data; or in the writing of the manuscript.

## Glossary and Abbreviations

| | |
|---|---|
| AI | artificial intelligence |
| as-built | the true physical system including any modifications made during its construction and/or operation as opposed to the nominal system during the design phase |
| binary plant | a power plant that transfers geothermal heat to a secondary fluid with a lower boiling point to drive a turbine generator |
| capacity factor | the ratio of actual power generated to the theoretical maximum output of a power station |
| digital twin | a digital representation of a physical system |
| forecast model | a relational data model that uses trained regressions to predict future operational states |
| GOOML | Geothermal Operational Optimization with Machine Learning |
| hindcast | a forecast model used for validation purposes that is based on some historical data such as known operator actions |
| historical model | a relational data model built using historical data |
| hybrid data-driven thermodynamics model | in the context of this work, GOOML is a hybrid model that relies heavily on traditional thermodynamics (e.g., fluid properties and conservation equations) but uses data-driven machine learning models to determine the behavior of the system within the thermodynamic operational space |
| IP | intermediate pressure |
| join junction | a component where two or more flows are joined to one |
| LP | low pressure |
| mass take | the total mass extracted by a geothermal system |
| MAE | mean absolute error calculated as: $\frac{\sum_{i=1}^{n} abs(x_G - x_T)}{n}$ where $x_G$ is the GOOML predicted value, $x_T$ is the true historical value, $n$ is the number of observations, and $abs$ is the absolute value operator |
| MBE | mean bias error calculated as: $\frac{\sum_{i=1}^{n} (x_G - x_T)}{n}$ where $x_G$ is the GOOML predicted value, $x_T$ is the true historical value, and $n$ is the number of observations |
| ML | machine learning |

| POI | Poihipi power station at the Wairakei Geothermal Field |
|---|---|
| RELU | rectified linear unit |
| separator/flash plant (FP) | a vessel that separates steam and liquid phases from a two-phase flow input, often involving pressure drop and cyclonic separation |
| split junction | a component where input flow is split into two distinct outputs, e.g., steam and liquid in the case of a separator |
| steamfield | a network of wells, pipelines, separators, turbine generators, and binary plants used to harness geothermal energy |
| TG | a turbine generator system that uses steam to generate electricity |
| THI | Te Mihi power station at the Wairakei Geothermal Field |
| TFT | tracer flow test—a method to assess energy and flow rate from a geothermal well |
| two-phase flow | a thermodynamic state of water where both saturated liquid and steam exist simultaneously |
| Willans Line | a highly simplified linear mass-to-power relationship used to represent turbine generator systems |
| WRK | Wairakei power station at the Wairakei Geothermal Field |
| WHS | well head separator—a small two-phase separator dedicated to a single well, typically mounted directly on the well head itself |

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
