# Peer review of "A New Modeling Framework for Geothermal Operational Optimization with Machine Learning (GOOML)"

_energies, doi:10.3390/en14206852_

Round 1

Reviewer 1 Report

Too many sentences are written in complex sentences. They are hard understand. Please break them up to simplify.

Why is there so many assumptions?

I put some comments on the manuscript please see them. Please send me a version with line numbers so that I can perform a detailed review.

Author Response

Please see the attachment which has the line numbers as requested by the reviewer. 

15 October 2021

Dear Mr. Ribac,

RE: Reviews of “A New Modeling Framework for Geothermal Operational Optimization with Machine Learning (GOOML)”

On behalf or my colleagues, I would like to thank you and the reviewers for the constructive criticism and suggested revisions of our submission. We have integrated several changes to the manuscript, and we believe that these changes have improved our submission.

Below, I attach a line-by-line rebuttal to the suggested changes and in the instance of not accepting a suggested change, have elaborated why we chose not to do so. The majority of the suggestions we have implemented, and I trust you will find the commentary below insightful. We have also uploaded a revised version of the manuscript

Thank you again for your consideration of our manuscript and we are eager to see it published very soon.

Yours sincerely,

Dr. Paul A. Siratovich (on behalf of my co-authors)

DIRECTOR

Upflow Limited

PO Box 61, Taupo 3330, New Zealand
MOBILE: +64 21 246 4931

E-MAIL: [email protected]

Reviewer 1 initial commentary:

Too many sentences are written in complex sentences. They are hard understand. Please break them up to simplify.

  • We have attempted to clarify the language used through improving the glossary/abbreviation list and hope the changes will aid readers in understanding.

Why is there so many assumptions?

  • We have added a statement to Section 2.4 to clarify why so many assumptions are made and are necessary in this manuscript.

I put some comments on the manuscript please see them. Please send me a version with line numbers so that I can perform a detailed review.

  • We provided the reviewer with an enumerated draft that was used to develop the commentary below.

Reviewer 1 secondary commentary:

“In the last review, I provided a broad review about the modeling framework. In this review, I emphasized more on writing, grammatical styles. First, I would like to thank the authors for clarifying a few terms and adding a glossary of several terms that are generally unknown to the general audience. Below, I provided several suggestions, which will enhance the quality of the manuscript.”

We greatly appreciate the suggested changes from the reviewer and have done our best to integrate them into the text as recommended albeit with some suggestions we would like to leave as written.

Line 17: the larger system => larger systems

  • We agree to the suggested change

Line 24: “as-built” not clear

  • Agreed and added to glossary

Lines 26-27: I would not claim it is generic. I would say GOOML provides good results in all case studies and can be applied to any site.

  • Modified this section: This modeling framework has already been applied to several geothermal power plants and has provided reasonably accurate results in all cases. Therefore, we expect that the GOOML framework can be applied to any geothermal power plant around the world.

Lines 37-38: Do we need “Opportunities for?”

  • We start the next sentence with opportunities for.... we have edited the sentence to end in "that enable geothermal as a baseload energy source.

Line 76: here: could be “provided below” because you are writing “First” as the first sentence.

  • Changed here: to here. to match the other First, Second, and Third sentence structures

Line 87: “Overarching these challenges” seems confusing

  • We feel that the term overarching captures the essence of what we are trying to convey with this statement and would prefer to leave unchanged.

Line 177: how conservation of mass and energy is maintained?

  • Agreed, modified the sentence: "Conservation of mass and energy is also maintained in the forecast system by constraints on the individual component regression models."

Figure 1: Require a description of FP.

  • Added more description of abbreviations and symbols in the Figure 1 caption.

Table 1: Turbine-Generator => Turbine Generator ?

  • Agreed we think no hyphen is more grammatically correct. Changed in all occurrences throughout the paper.

Figure 7: Do we need the right figure for the Te Mihi system?

  • Yes, we think the confusion lies in the three power stations at the Wairakei geothermal complex (WRK, THI, and POI). We have added a note in the text and further defined the three abbreviations in the nomenclature table.

Reviewer 2 Report

The paper entitled “A New Modeling Framework for Geothermal Operational Optimization with Machine Learning (GOOML)” was carefully reviewed. In general, this is an interesting topic and a well-written and well-organized paper. To improve its quality, the following comments are suggested:

- Please add the “Abbreviations” to the paper.

- Please specify “WHS” and “FP” in Fig. 1 in its caption.

- Please specify the symbols used (only main parameters) in Fig. 1 in its caption.

- Please add the “Nomenclature” to the paper.

- There are some minor grammatical errors in the manuscript. Please revise them.

Author Response

15 October 2021

Dear Mr. Ribac,

RE: Reviews of “A New Modeling Framework for Geothermal Operational Optimization with Machine Learning (GOOML)”

On behalf or my colleagues, I would like to thank you and the reviewers for the constructive criticism and suggested revisions of our submission. We have integrated several changes to the manuscript, and we believe that these changes have improved our submission.

Below, I attach a line-by-line rebuttal to the suggested changes and in the instance of not accepting a suggested change, have elaborated why we chose not to do so. The majority of the suggestions we have implemented, and I trust you will find the commentary below insightful. We have also uploaded a revised version of the manuscript

Thank you again for your consideration of our manuscript and we are eager to see it published very soon.

Yours sincerely,

Dr. Paul A.  Siratovich (on behalf of my co-authors)

DIRECTOR

Upflow Limited

PO Box 61, Taupo 3330, New Zealand
MOBILE: +64 21 246 4931 

E-MAIL: [email protected]

Reviewer 2 commentary:

The paper entitled “A New Modeling Framework for Geothermal Operational Optimization with Machine Learning (GOOML)” was carefully reviewed. In general, this is an interesting topic and a well-written and well-organized paper. To improve its quality, the following comments are suggested:

Please add the “Abbreviations” to the paper.

  • We have integrated the abbreviations into a combined glossary and abbreviations table.

Please specify “WHS” and “FP” in Fig. 1 in its caption.

  • We have addressed this in the caption as requested by both reviewers.

Please specify the symbols used (only main parameters) in Fig. 1 in its caption.

  • As above have specified the desired changes in the caption.

Please add the “Nomenclature” to the paper.

  • We have updated the glossary/abbreviations to capture definitions of items.

There are some minor grammatical errors in the manuscript. Please revise them.

  • We have integrated this feedback and revised to capture any outstanding errors.
